# Effects of Drought Stress on Leaf Functional Traits and Biomass Characteristics of *Atriplex canescens*

**DOI:** 10.3390/plants13142006

**Published:** 2024-07-22

**Authors:** Shuai Wang, Hai Zhou, Zhibin He, Dengke Ma, Weihao Sun, Xingzhi Xu, Quanyan Tian

**Affiliations:** 1Linze Inland River Basin Research Station, Chinese Ecosystem Research Network, Northwest Institute of Eco-Environment and Resources, Chinese Academy of Sciences, Lanzhou 730000, China; wangs0504@163.com (S.W.); zhouhai1201@lzb.ac.cn (H.Z.); madengke@nieer.ac.cn (D.M.); sunweihao1996@163.com (W.S.); tianquany@126.com (Q.T.); 2University of Chinese Academy of Sciences, Beijing 100049, China; 3College of Pratacultural Science, Gansu Agricultural University, Lanzhou 730070, China; xxz20001019@163.com

**Keywords:** *A. canescens*, leaf biochemical traits, leaf morphological traits, biomass partitioning patterns, adaption strategies, tradeoffs

## Abstract

Drought is a critical factor constraining plant growth in arid regions. However, the performance and adaptive mechanism of *Atriplex canescens* (*A. canescens*) under drought stress remain unclear. Hence, a three-year experiment with three drought gradients was performed in a common garden, and the leaf functional traits, biomass and biomass partitioning patterns of *A. canescens* were investigated. The results showed that drought stress had significant effects on *A. canescens* leaf functional traits. *A. canescens* maintained the content of malondialdehyde (MDA) and the activity of superoxide dismutase (SOD), but the peroxidase (POD) and catalase (CAT) activity decreased, and the content of proline (Pro) and soluble sugar (SS) increased only under heavy drought stress. Under drought stress, the leaves became smaller but denser, the specific leaf area (SLA) decreased, but the dry matter content (LDMC) maintained stability. Total biomass decreased 60% to 1758 g under heavy drought stress and the seed and leaf biomass was only 10% and 20% of non-stress group, but there had no significant difference on root biomass. More biomass was allocated to root under drought stress. The root biomass allocation ratio was doubled from 9.62% to 19.81% under heavy drought, and the root/shoot ratio (R/S) increased from 0.11 to 0.25. The MDA was significantly and negatively correlated with biomass, while the SPAD was significantly and positively correlated with total and aboveground organs biomass. The POD, CAT, Pro and SS had significant correlations with root and seed allocation ratio. The leaf morphological traits related to leaf shape and weight had significant correlations with total and aboveground biomass and biomass allocation. Our study demonstrated that under drought stress, *A. canescens* made tradeoffs between growth potential and drought tolerance and evolved with a conservative strategy. These findings provide more information for an in-depth understanding of the adaption strategies of *A. canescens* to drought stress and provide potential guidance for planting and sustainable management of *A. canescens* in arid and semi-arid regions.

## 1. Introduction

Plants have evolved different strategies to reduce the adverse effects of environmental stress [1,2], such as adjust morphological, physiological and biochemical characteristics at the whole-plant, organ and tissue levels [3]. Drought is one of the key ecological factors that constrained plant growth and performance in arid and semi-arid area [4,5]. Considering more severe and frequent drought events to be even more pronounced in the context of climate change [6], it is urgent to understand how plants respond to drought stress. Species in water-limited environments may operate a variety of tradeoffs between various functional traits to cope with resource stress [7]. Functional traits reveal the relations between plants and environment, and objectively reflect the adaptability of plants to the environment and the utilization of resources [8].

Among the functional traits of plant organs, leaf functional traits are the decisive factors of plant physiological and biochemical cycle [9]. The leaf is the basic organ of photosynthesis, and plays a decisive role to plant productivity [10]. Leaf has the largest contact area with environment, and it is one of the most sensitive and vulnerable organs that has strong phenotypic plasticity [11]. Thus, leaf functional traits are excellent indicators to reflect the ability of resources acquisition and utilization, as well as survival strategies of plants under environmental changes [12,13,14]. Leaf functional traits can be divided into physiological, biochemical and morphological traits [15,16,17], leaf physiological traits are those related to photosynthesis, biochemical traits are those related to enzyme activity and substance content, and the leaf morphological traits are the characteristic of shape, weight and water content. Plants have been identified that accumulate a certain quantity of reactive oxygen species (ROS) under drought stress [18], and thus stimulate antioxidant enzymes activity to removal ROS to ameliorate the damage [19]. The change of osmotic adjustment substances content are also key indicators to depict the adaptive mechanism of plants under drought stress [20]. The indicators related to leaf morphology are easy to measure and usually have good data quality [21], as well as being compared between different species, so they are widely used to evaluate the adaptability of plants to the harsh environment [22]. Previous study has shown that leaf morphological traits associated with stress resistance has the expense of those associated with rapid resource capture and growth [23]; this means the plants sacrificed their rapid growth and selected conservative strategies to adapt to limited resources.

Besides leaf functional traits, biomass and biomass partitioning patterns are overriding traits when express the stress resistance of plants [24]. Variation in plant size or total biomass reflects the resources acquisition capability and biomass production. Biomass allocation to different parts determines the fitness and reproductive success; it also reflects the adaptability to heterogeneous environments [25]. The major theory in biomass partitioning is optimal partitioning theory, which indicates that plants allocate more biomass to the organs by capturing the scarcest resources (e.g., water, nutrients, or light) to optimize plant performance and maximize fitness [5]. Several studies have shown that plants invest more biomass to roots to enhance water uptake when water is limited [26].

*Atriplex canescens* is a drought-tolerant shrub that shows strong stress resistance to harsh environments. *A. canescens* is native to North America; due to its excellent ecological and forage values, it has been widely introduced to arid and semi-arid regions all around the world [27]. In 1986, *A. canescens* was introduced into China, and it has been used to control soil erosion, restore degraded grassland, combat desertification and as feed [28]. In the last decade, *A. canescens* has been widely used in railway and highway side slope protection and mining site ecological restoration. *A. canescens* grows well in high-altitude areas of Qinghai Tibet Plateau at 4200 m (Cuomei County, Tibet autonomous region), and its ecological properties and feed utilization are highly valued by local governments in certain ultrahigh-altitude regions. Nowadays, the most important utilization of *A. canescens* in China is as a host of *Cistanche deserticola*, which is a traditional Chinese medicinal material [29]. Because of its high economic value, it has been widely planted in North and Northwest China. In this region, the annual precipitation is from 35 mm (hotan county, Taklimakan Desert) to 450 mm (Jingbian county, south of Mu Us Desert), and most of the planting area has less than 150 mm precipitation. In addition, the planting area in heavy drought regions continues to increase. So, understanding the performance of *A. canescens* under drought stress has great significance for planting and management under global warming conditions. Although many studies have discussed the drought resistance of *A. canescens* [30,31,32,33], most of the previous studies focused on potted seedlings, with only a few studies based on common garden experiments and mature individuals [34,35]. In addition, there are no comprehensive studies that combine leaf functional traits and biomass partitioning patterns.

The overall goal of this study was to clarify how *A. canescens* adapted to drought stress. *A. canescens* seedlings were transplanted to a common garden in 2021, and heavy drought, feeble drought, and non-drought stress conditions were simulated from then until sample collection and analysis, which were conducted in 2023. Our objectives were to (1) characterize leaf functional traits changes related to drought stress, (2) assess biomass variation and biomass partitioning patterns and (3) reveal the tradeoff strategies of *A. canescens* under drought stress. We expected to have a better understanding of the adaptability of *A. canescens* to drought, and to provide management experience for the planting and utilization of *A. canescens* in different area under climate change. 

## 2. Results

### 2.1. Leaf Biochemical Traits

It was observed that drought stress did not affect the MDA content significantly in the florescence or seed stage (Figure 1A). Heavy drought stress significantly decreased the SPAD in both stages, but feeble drought did not affect the SPAD (Figure 1B).

At the florescence stage, the SOD activity was almost steady under drought stress (Figure 2A) and there was also no significant difference at the seed stage. At both florescence and seed stages, the leaf POD activity of *A. canescens* from the heavy drought group was less than half of that of the non-stress group; the difference was significant. The POD activity of *A. canescens* from the feeble drought group was significantly lower than in the non-stress group at florescence stage, but not significantly at seed stage. The CAT activity showed the same pattern as the POD at florescence stage but was inconsistent at the seed stage. At the seed stage, the CAT activity of *A. canescens* under heavy and feeble drought were both significantly lower than in the non-stress group. At the florescence stage, both the Pro and SS contents showed the same pattern in that the value of the heavy drought group was significantly higher than that of non-stress group (Figure 2D,E). At the seed stage, the Pro of the heavy drought group showed the same pattern as that of the florescence stage, but there was no significant difference in SS. Feeble drought stress did not affect the Pro and SS in the florescence or seed stage.

### 2.2. Leaf Morphological Traits

The indicators related to leaf morphological traits are shown in Table 1. Drought affected leaf morphological traits significantly. At the florescence stage, there were 9 indicators that displayed a significant difference between the heavy drought and non-stress groups among the 13 measured indicators. Heavy drought decreased leaf length, leaf width, leaf area, leaf shape index, fresh weight, saturated fresh weight, dry weight and specific leaf area significantly, while also increasing leaf density significantly. Leaf thickness, leaf dry matter content, saturated water content and water content had not significantly changed. Most of the indicators of the feeble drought group changed with the same pattern as the heavy drought group, where only LL, LA, SLA and LD had a significant difference. At the seed stage, the general changes in leaf morphological traits under drought gradient were consistent with the florescence stage, but the difference in FW and SFW between the feeble drought group and non-stress group were significant and there were no significant changes in SLA or LD.

### 2.3. Biomass Allocation Variables

Biomass decreased significantly in response to drought stress (Figure 3A). The *A. canescens* from non-stress group had a total biomass of 4419 g, while the heavy drought group had total biomass of 1758 g, only 39.79% of that of the non-stress group. The *A. canescens* under feeble drought had a total biomass of 2935 g, which was significantly lower than that of the non-stress group but significantly higher than that of the heavy drought group. The biomass of different organs showed the same features as those of total biomass. The *A. canescens* from the heavy drought group had the lowest seed, leaf, stem and root biomass; the non-stress group had the highest biomass in those organs. All of the organs’ biomass under drought stress showed a significant difference, except for that of the root. The root biomass of *A. canescens* in the heavy, feeble and non-stress group was 348 g, 397 g and 425 g, respectively. The root biomass of the heavy drought group was higher than seed and leaf biomass but lower than the stem biomass, while in the feeble drought group, the root biomass was higher than the seed biomass but lower than leaf and stem biomass. The root biomass of the non-stress group was lower than that of other organs. The *A. canescens* from the heavy drought group had the lowest reproductive biomass of 67 g, the feeble drought group had a median value of 225 g and the non-stress group had the highest value of 551 g. Under drought stress, reproductive organs had the lowest biomass compared to the other organs, but in the non-stress group, it was lower than that of the leaf and stem but higher than that of the root.

Significant changes in biomass allocation were observed in response to drought stress (Figure 3B). The *A. canescens* under no drought stress allocated more biomass to the leaves and seeds, while under drought stress, more biomass was allocated to the roots and stem. Under heavy drought, *A. canescens* allocated 19.81% of total biomass to the roots, which was significantly higher than that of the non-stress group (9.62%) and feeble drought group (13.53%). There was no significant difference between the feeble drought and non-stress groups in root biomass allocation ratio. *A. canescens* allocated 12.34% of its total biomass to the leaves under heavy drought conditions, which was significantly lower than that of the non-stress group (20.60%). The *A. canescens* under feeble drought conditions had a leaf allocation ratio of 17.81%, which is not a significant difference compared to the non-stress group and heavy drought group. Under heavy drought conditions, the reproductive organ had only 3.82% of total biomass, that in the feeble drought group had 7.67% and that in the non-stress group had 12.49%; there were significant differences among drought gradients. *A. canescens* allocated most of the biomass to the stem (all groups above 57%); as drought stress strengthened, the stem biomass allocation ratio increased. Under heavy drought conditions, the stem allocation ratio was as high as 64.03%, significantly higher than in the non-stress group. Drought affected the root/shoot ratio of *A. canescens*: the heavy drought group had an R/S of 0.25, the feeble drought group had 0.16 and the non-stress group had 0.11. The R/S of *A. canescens* under heavy drought was significantly higher than those of the non-stress and feeble drought stress groups.

### 2.4. Correlations between Leaf Functional Traits and Biomass

At the florescence stage, the MDA content was negatively correlated with the biomass of the root, stem, leaf, seed and the total, and both the POD and CAT activity were positively correlated with seed ratio and negatively correlated with root ratio, while the Pro and SS were negatively correlated with seed ratio and positively correlated with root ratio (Figure 4). The SS was also negatively correlated with stem and total biomass and positively correlated with R/S. SPAD was positively correlated to all biomass indicators and leaf ratio, while it has a negatively correlation with stem ratio and R/S. At the seed stage, the POD was positively correlated with seed ratio and negatively correlated with root ratio, while the Pro was negatively correlated with biomass, leaf and seed allocation ratio and positively correlated with root allocation ratio and R/S. Most of the correlations reached a significant level. The SPAD had a positive correlation with biomass indicators and leaf and seed allocation ratio, and had negative correlations with root and stem allocation ratio and R/S.

At the florescence stage, the indicators related to leaf shape and weight were positively correlated with biomass and leaf and seed allocation ratio, and negatively correlated with root and stem allocation ratio and R/S (Figure 5). The LD was positively correlated with stem allocation ratio and R/S and negatively correlated with leaf, seed and total biomass. The SLA was negatively correlated with stem allocation ratio. There were no significant correlations between leaf morphological traits and root biomass, and only SFW and DW showed significant negative correlations with root biomass allocation ratio. The correlations at seed stage showed same pattern as that of the florescence stage, but SLA had significant correlations with stem, leaf and total biomass and leaf allocation ratio.

## 3. Discussion

Plant functional traits are the changes in the morphological structure and physiological and biochemical characteristics of plants in the long process of evolution. These changes can objectively reflect the response strategies of plants to the environment [36]. Plant functional traits do not respond to the environment in isolation but interact and correlate with each other. In the process of long-term adaptation to the environment, they coordinate and balance each other by adjusting different traits, so as to improve the adaptability to the environment [37]. Plants are subjected to drought and have evolved multi-mechanisms to minimize the negative effects [38]. This study suggests that *A. canescens* changes its leaf functional traits, biomass and their allocation characteristics in response to drought stress.

### 3.1. Leaf Biochemical Response to Drought Stress

Drought stress causes the excessive accumulation of reactive oxygen species (ROS), which causes the biomembrane lipid peroxidation reaction [39]. The toxicity of lipid peroxidation products induces damage to membrane structure and function, and then affects the normal function of the leaf [40]. MDA is the primary product of lipid peroxidation; the content of MDA can be used as an indicator to reflect the degree of membrane damage and the self-recovering ability in plants [41]. Other types of ROS damage caused are photosynthetic pigment degradation and decreased chlorophyll content [42], which affect the activity of photosynthetic enzymes and the efficiency of photosynthesis, and inhibit plant growth and development [43]. SPAD reflects the relative content of chlorophyll; it is an important indicator for revealing plant photosynthesis ability [44]. In our study, the MDA content showed no significant difference but the SPAD decreased significantly under heavy drought. These changes mean that the long drought stress did not damage the biomembrane of *A. canescens* significantly but significantly affected its photosynthetic system, thus potentially hampering photosynthetic potential [45], and as a result, it caused a biomass decrease in *A. canescens*.

Plants usually enhance their drought resistance by increasing the expression of protective enzymes to eliminate ROS under drought stress [46]. SOD, POD and CAT are the most important enzymes in the plant antioxidant defense system. Under drought stress, SOD, POD and CAT react synergistically, retain active oxygen metabolism balance and protect biomembrane structure and function [47]. SOD can catalyze the superoxide radical reaction; the generated peroxides can be decomposed into water and oxygen with the catalyst of POD and CAT [48]. Therefore, the activity of these enzymes plays an important role in reflecting drought resistance ability. In our study, the activity of SOD had no significant difference, but the activity of POD and CAT decreased significantly under heavy drought. This indicated that under drought stress, *A. canescens* has high superoxide disposal efficiency, thus why there was no excessive accumulation of MDA, but drought stress significantly decreased the peroxide disposal ability, which caused the accumulation of peroxide and damaged plants growth. Osmotic regulation is another biochemical mechanism for plants to adapt to arid environments. An accumulation of osmotic adjustment substances can reduce osmotic potential to maintain water absorption capacity from drought soil and maintain normal physiological and biochemical functions [49]. Pro and SS are the most studied osmotic adjustment substances [50]. Under drought stress, the content of SS and Pro increased, which means *A. canescens* accumulated a mass of osmotic adjustment substances to mitigate the negative effect of drought. 

At the seed stage, there was a relatively higher value of leaf biochemical traits. This is probably because the experiment at seed stage was conducted in early Oct.; at that time, the temperature in the common garden was low (the night temperature was near 5 °C), and the *A. canescens* suffered from the combined stress of drought and low temperature. Under the dual stress, *A. canescens* further improved the activity of SOD and increased the content of osmotic adjustment substances to alleviate the impact of stress. The increased osmotic adjustment substances can lower the freezing point to prevent frost damage [46]. Although the low temperature in the seed stage increased the antioxidant enzyme activity and the content of osmotic substances, the difference in those indicators under drought gradients showed the same pattern at the florescence stage. Therefore, the results of the seed stage further verified the biochemical adaptation mechanism of *A. canescens* to drought stress. Under feeble drought stress, the activity of enzymes and the content of osmotic adjustment substances had similar overall change trends to the heavy drought group, but the content of SPAD in both stages, the activity of POD in seed stage and the activity of Pro in florescence stage showed no significant difference to the non-stress group. This may indicate that *A. canescens* has a relatively strong adaptability to feeble stress. 

### 3.2. Leaf Morphological Response to Drought Stress

As the main production organ of plants, leaves have strong plasticity and are closely related to resource acquisition and utilization. The leaf is the key entry point for studying the relationship between plants and the environment [51]. Under drought stress, leaves will make a tradeoff in functional traits between fast growth and stress tolerance [52,53]. Under drought stress, the leaf of *A. canescens* becomes smaller but denser. These changes reflect that under drought stress, *A. canescens* makes tradeoffs and follows a series of conservative strategies to adapt to drought stress. In general, larger leaf area can increase the light absorption surface and improve photosynthetic capacity, which promote the fast growth of plants. However, a larger leaf area usually enhances the respiration and transpiration cost, and plants tend to decrease their leaf area to reduce the transpiration in arid regions [54]. SLA and LDMC are frequently used to reflect the resource utilization efficiency of plants [55]. Plants with high SLA are more likely to use a resource acquisition strategy, which promotes the growth rate in plants [56], while plants with lower SLA are more likely to use a stress tolerance strategy, which forms small but heavy leaves and shows strong resistance to drought stress [57]. Higher LDMC means plants accumulate more substances in leaves, which increases the diffusion distance or resistance of water inside the leaves to the surface, and reduces the water loss inside the plant to mitigate adverse environmental effects [58]. Water is limited under heavy drought, but there is no significant difference in LDMC, SWC or WC, and the LA and SLA decrease, which means that *A. canescens* retains high water absorption ability and decreased leaf area to reduce water transpiration to guarantee the necessary water content, which is a key factor for the survival and growth of plants [6].

### 3.3. Biomass Allocation Response to Drought Stress

Biomass is an important indicator to evaluate the growth status and productivity levels of plants [59]. Plants adjust their biomass and biomass allocation to adapt to environmental changes [60]. In this study, the biomass of different organs showed that *A. canescens* obtained a conservative strategy to coping with drought. The total biomass under heavy drought and feeble drought was only 40% and 67% of that of the non-stress group. Among all of the organs, the most decreased were seeds and leaves, while the least decreased was the root. The root system is the main organ for obtaining resources, as plants absorb water and nutrients through their massive root systems [61]. The stem transports water and nutrients from the root to other parts and provides architectural support for leaves and reproductive organs [5]. The leaf is a pivotal component in photosynthesis and also a major outlet of water via transpiration [5]. Under heavy drought, the seed and leaf biomass were only 10% and 20% of that of non-stress group, and the root biomass was over 80% of that of non-stress group. This demonstrated that *A. canescens* allocates high priority in guaranteeing root investment under drought stress. Even when the total biomass sharply decreased, the root biomass was relatively stable and remained at a high level, which ensured the water absorption capacity in the arid environment. Meanwhile, *A. canescens* invested less in its leaves to reduce the water loss, which is a tradeoff between rapid growth potential and survival ratio [52]. The investment in offspring was the most affected, which indicates *A. canescens* sacrificed its fecundity to guarantee the growth of its parental generation. Biomass allocation can also reflect the conservative strategy of *A. canescens* under drought. The organ biomass allocation ratio and root/shoot ratio showed that under drought stress, *A. canescens* allocated more biomass to the root system and less to the leaves, which dissipated the water more slowly. 

Previous studies have shown two classical theories on biomass allocation: optimal partitioning theory and allometric partitioning theory [62]. According to optimal partitioning theory, plants give priority to allocate more biomass to organs that can obtain more restricted resources to ensure growth [63]. Thus, the organ allocation ratio is the response of plants to resource availability [64]. Allometric partitioning theory claims that the allocation ratio in different organs is a power function of plant size independently of environmental variation [65]. In our study, the biomass allocation patterns of *A. canescens* under drought stress were conformed with optimal partitioning theory. As a result of water limitation, *A. canescens* allocated more biomass to its roots to obtain water, and decreased its leaf biomass ratio to reduce transpiration. In general, under drought stress, *A. canescens* takes a conservative resource allocation strategy, it allocates more resources to the organs that best maintain restricted resources and reduces the investment in organs that consume those limited resources. This conclusion is consistent with a large number of studies, which shows that under drought stress, plants allocate more resources to their roots and reduce their leaf and reproductive investment [5,25,26].

### 3.4. The Relationship between Leaf Functional Traits and Biomass

The relationships between plant functional traits are ubiquitous [66]. Biomass is the result of photosynthate distribution and material accumulation. As the main organs of photosynthesis in plants, leaf functional traits have a conjugate relationship with biomass and biomass partitioning patterns [67]. Study on these relationships hold significance for the quantification and summary of the tradeoff and resource utilization strategies in plants under drought stress [8]. The florescence stage is one of the peak growth periods and the main stage of biomass accumulation, while at seed maturation stage, growth stagnates [68]. Thus, this study only discussed the correlations between leaf functional traits and biomass at florescence stage. Actually, the correlations at seed stage showed the same pattern as the florescence stage. In this study, the MDA content was negatively correlated with all of the biomass indicators, while SPAD was positively correlated with the total biomass and the aboveground organs biomass. It was also shown that there was a significantly negative correlation with MDA and SPAD, consistent with previous studies [69,70]. This indicated that SPAD is highly correlated with MDA content and biomass; therefore, the SPAD can be used to reflect the leaf damage degree and plant growth status of *A. canescens* under drought stress. Two antioxidant enzymes, POD and CAT, showed significant correlations with root and seed allocation ratio, and osmotic adjustment substance, and Pro and SS were also significantly correlated with the root and seed allocation ratio, but the positive or negative correlations were contrary to POD and CAT. It was also shown that there were negative correlations of POD and CAT with Pro and SS, which means that under drought stress, the osmotic adjustment substance increased to maintain the normal physiological and biochemical activity when the activity of POD and CAT has been inhibited [69,71]. There may have a synergistic effect between POD and CAT and also in Pro and SS [72]. In addition, there shown a complementary effect of enzymes activity and osmotic adjustment substance of *A. canescens* under drought stress.

In our study, the leaf morphological traits related to shape and weight had significant correlations with total and aboveground organ biomass; stem, leaf and seed allocation ratio; and R/S, which means that leaf shape and weight traits can explain the biomass and biomass partitioning pattern, and these results were consistent with previous studies [57,73,74]. It should be noted that there was an extremely negative correlation between SLA and LDMC, which was consistent with the study of Pontes et al. [75]. Garnier et al. [76] and Elger et al. [77] hold that SLA and LDMC do not directly affect productivity in plants but are related to other functional features such as palatability and litter decomposition characteristics. The synergism, complementation and trade-off among traits is one of the important mechanisms of population dynamics and community construction [66,78]. Through this research on the correlations among functional traits, traits that are highly correlated with others and easy to measure as key indicators can be screened out in future research, and this will make the research more focused and efficient. In this study, under drought stress, leaf MDA and SS content and SPAD of *A. canescens* were significantly correlated with each other and also had significant correlations with biomass and biomass partitioning, so more attention should be paid to these indicators in related future studies.

## 4. Materials and Methods

### 4.1. Experimental Site

The experiment was conducted in the Linze Inland River Basin Research Station of the Chinese Academy of Sciences (39°21′ N, 100°07′ E), which is located in the middle of Hexi Corridor of Gansu Province, on the south edge of the Badain Jaran Desert. The altitude of the experimental site was 1367 m. The soil was classified as gray-brown desert soil, which has high sand content, low organic matter content, poor water holding capacity and a lack of nutrients [79]. The experimental site has a typical arid desert climate with hot dry summers and cold winters. The average annual temperature is 7.6 °C. The average annual precipitation is 116.8 mm with 80% occurring from June to September [80]. The average annual open water (pan) evaporation is 2388 mm, which is 20-fold the annual precipitation. The number of frost-free days is between 150 and 160; the first frost day is usually in early or mid-October.

### 4.2. Experimental Design

The experimental site was divided into 3 fields, each field with a total area of 450 m^2^ (30 m × 15 m). The fields were isolated by a 60 cm × 40 cm (width × height) soil bund. The groundwater depth of the fields was about 4.5 m. One-year-old *A. canescens* seedlings were transplanted in rows of 1.5 m × 3 m in the spring of 2021. After transplantation, the experimental fields were irrigated every 5–10 days depending on the soil moisture to facilitate plant establishment. Groundwater was used for irrigation in this study, and flood irrigation was applied twice, with an irrigation amount of 1200 m^3^ hm^−1^ each time. After successful establishment, simulated drought gradients were applied by controlling the irrigation frequency. The south field was the non-stress group, which was irrigated monthly from April to October each year; the middle filed was the feeble drought stress group, which was irrigated bi-monthly (April, June, August and October) each year. Irrigation dates were determined according to local weather conditions, usually in the middle ten days of the month. The irrigation amount was 900 m^3^ hm^−1^ per application, which was equal to 90 mm precipitation. The north field was the heavy drought stress group, which was never irrigated, and the only water source was precipitation. The precipitation amounts of the experimental site were 108 mm, 91 mm and 73 mm in 2021, 2022 and 2023, respectively. Soil water content of 0–150 cm soil layers was measured by oven drying method at the florescence and seed maturation stage (Figure 6). During the experiment, no fertilizer was applied in the field, the weeds were mowed manually once a year, and no other management practices were applied in the common garden.

### 4.3. Data Collection and Processing

#### 4.3.1. Leaf Functional Traits

Leaf biochemical traits were measured at florescence (late June) and seed maturation stage (early October). The enzyme activities of superoxide dismutase (SOD), catalase (CAT) and peroxidase (POD) and the content of malondialdehyde (MDA), proline (Pro) and soluble sugar (SS) were determined using biochemical kits (G0101W, G0105W, G0107W, G0109W, G0111W and G0501W, respectively) [81], which were purchased from Suzhou Grace Biotechnology Co., Ltd (Suzhou, Jiangsu, China). The indicators were tested following the manufacturer’s instructions. All indicator tests were repeated three times. Relative chlorophyll content (SPAD) was measured by SPAD-502 chlorophyll meter (Konica Minolta, Osaka, Japan) [82]. The average value of four data collected in a tree was treated as a sample datum, and ten trees were randomly selected and recorded in each group.

Leaf morphological traits were also measured at both stages. The indicators were tested on the day after the biochemical traits were tested. Ten fully developed and undamaged leaves from one tree were treated as a sample; seven samples from seven individual trees were obtained in each group. The fresh weight (FW) was determined with an analytical balance. Leaf thickness (LT) was recorded with a digital vernier caliper. The leaves were scanned with a scanner (Epson Perfection V850 Pro, Seiko Epson Corporation, Japan). The leaf length (LL), leaf width (LW) and leaf area (LA) were analyzed with Image J software (Version 1.8.0, National Institute of Health, America) [7]. The leaves were soaked in distilled water for 24 h in darkness, then patted dry on sterile filter paper, and their saturated fresh weight (SFW) was determined [31]. The leaves were dried to constant weight in an oven at 70 °C and their dry weight (DW) was determined. Other leaf morphological traits were calculated using the above data, which were specific leaf area (SLA = leaf area/dry weight), leaf shape index (LI = leaf length/leaf width), leaf density (LD = dry weight/(leaf area × leaf thickness)), dry matter content (LDMC = dry weight/saturated fresh weight), saturated water content (SWC = (saturated fresh weight − dry weight)/saturated fresh weight) and water content (WC = (fresh weight − dry weight)/fresh weight) [24,75].

#### 4.3.2. Biomass Partitioning Patterns

Whole plants were harvested (3 in plants each group) in early October 2023 [83]. Plants were separated into four components: (1) root, (2) stem, (3) leaf and (4) seed. All of the roots, leaves and seeds were placed in paper bags and dried to a constant weight in an oven at 70 °C. All fresh stems were weighed, and a roughly 500–800 g sample was selected for drying to constant weight to evaluate the stem biomass. Total aboveground biomass (including stem, leaf and seed) was calculated as shoot biomass; the reproductive biomass was the seed biomass. From these data, we calculated the biomass allocation ratio of each organ, and the root/shoot ratio (R/S) [84].

### 4.4. Data Analyses

All statistical analyses were conducted with SPSS software (Version 27.0.1, SPSS Inc. Chicago, IL, USA). One-way ANOVA was applied, and then an LSD test was used to detect differences among groups in leaf functional traits, biomass, and biomass allocation; the statistical significance level was set at *p* ≤ 0.05. Correlations between leaf functional traits and biomass at different growth stages were separated, determined using the Pearson coefficient. All the figures were drawn with Origin Software (Version 2021, Origin Lab Crop. Northampton, MA, USA).

## 5. Conclusions

Our common garden experiment results found that the biochemical and morphological traits of leaves, as well as the biomass and biomass allocation of *A. canescens*, significantly changed under drought stress. The POD and CAT activity decreased, the content of osmotic adjustment substance increased, and the content of MDA and activity of SOD were maintained. Under drought stress, leaves became smaller but denser and maintained high water content and stable dry matter content. Drought stress significantly decreased the total biomass and all organ biomasses except that of the root. Furthermore, the biomass allocated to leaves and reproductive organs decreased, the root allocation rate increased, and the root shoot ratio increased. In general, under drought stress, *A. canescens* maintains survival through biochemical regulation and make a tradeoff between rapid growth and drought resistance. This assessment was made based on leaf morphology, biomass and biomass allocation, showing that a conservative survival strategy had been adapted. Specifically, *A. canescens* increased its investment in its roots to absorb the limited water and reduced its investment in leaves and offspring to reduce the consumption of water, in order to adapt to the arid environment. The results of this study provide valuable insights to how *A. canescens* respond and adapt to drought stress in natural environments. However, as the adaption mechanism of plants to drought stress involves complex physiological processes, photosynthetic, plant hydraulic and gas exchange characteristics need further studies to understand the complete picture of how *A. canescens* adapt to drought stress.

## Figures and Tables

**Figure 1 plants-13-02006-f001:**
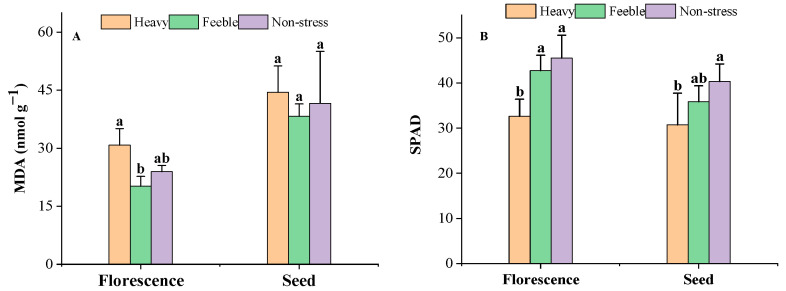
The MDA content (**A**) and SPAD (**B**) of *A. canescens* under drought stress. MDA: malondialdehyde. SPAD: relative chlorophyll content. Heavy means heavy drought stress; feeble means feeble drought stress; non-stress means no drought stress. Florescence or seed mean at florescence or seed stage. Different letters above the error bar (a, b) in the same indicators at the same stage means a significant difference (*p* ≤ 0.05). The same below.

**Figure 2 plants-13-02006-f002:**
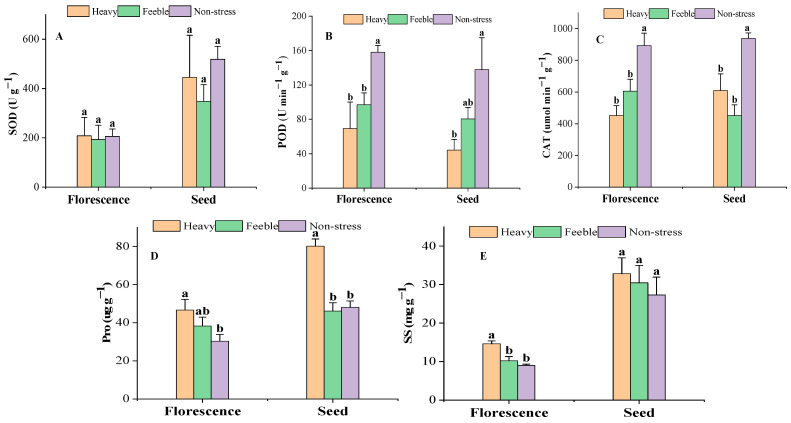
The activity of antioxidant enzymes and osmotic adjustment substance content under drought stress. (**A**) SOD: superoxide dismutase; (**B**) POD: peroxidase; (**C**) CAT: catalase; (**D**) Pro: proline; (**E**) SS: soluble sugar. Different letters above the error bar (a, b) in the same indicators at the same stage mean a significant difference (*p* ≤ 0.05).

**Figure 3 plants-13-02006-f003:**
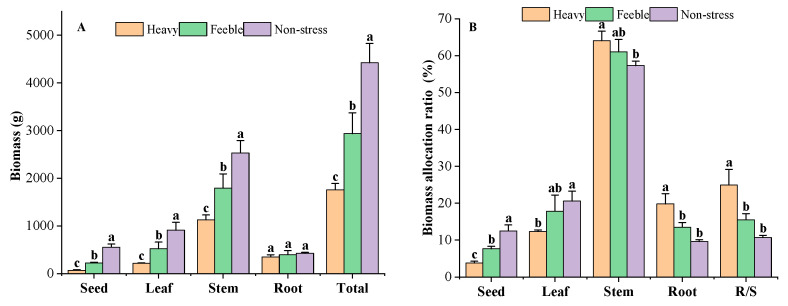
The biomass (**A**) and biomass allocation ratio (**B**) under drought stress. Total means the biomass of all the organs; R/S means root/shoot ratio. Different letters above the error bar (a, b and c) in the same indicators at the same stage mean a significant difference (*p* ≤ 0.05).

**Figure 4 plants-13-02006-f004:**
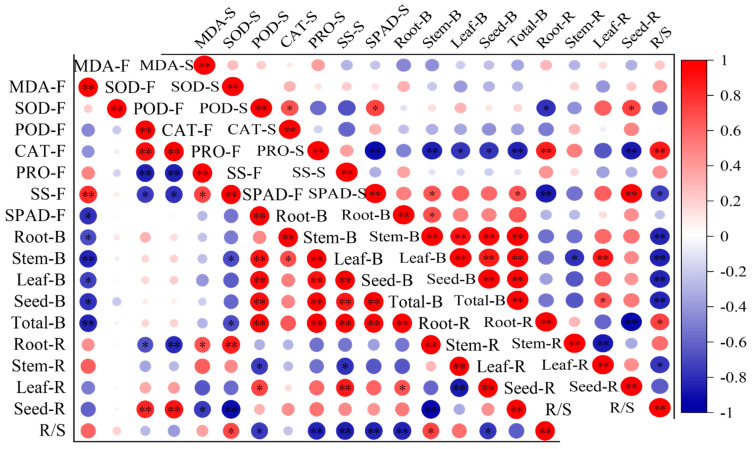
The correlations between leaf biochemical traits, biomass and biomass allocation ratio. The mean of the indicators is as the same in Figure 1, Figure 2 and Figure 3. The F after the leaf biochemical traits means at florescence stage; the S after leaf biochemical traits means at seed stage. The B after organs means biomass. The R after organs means biomass allocation ratio. * Means correlation significant at the 0.05 level (*p* ≤ 0.05); ** means correlation significant at the 0.01 level (*p* < 0.01). The same below.

**Figure 5 plants-13-02006-f005:**
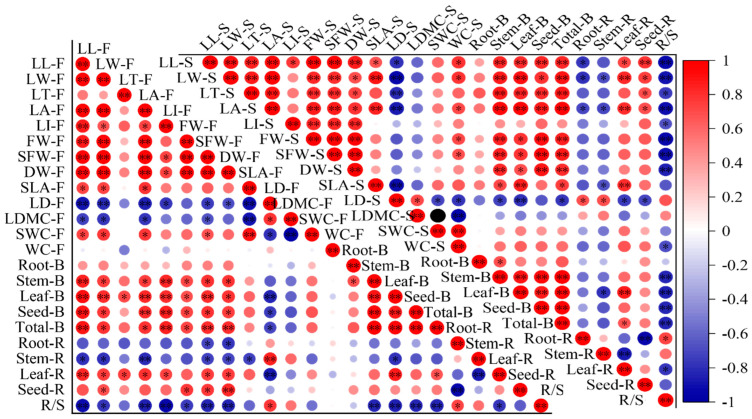
The correlations between leaf morphological traits, biomass and biomass allocation ratio. The mean of the indicators is as the same in Figure 3 and Table 1. * Means correlation significant at the 0.05 level (*p* ≤ 0.05); ** means correlation significant at the 0.01 level (*p* < 0.01).

**Figure 6 plants-13-02006-f006:**
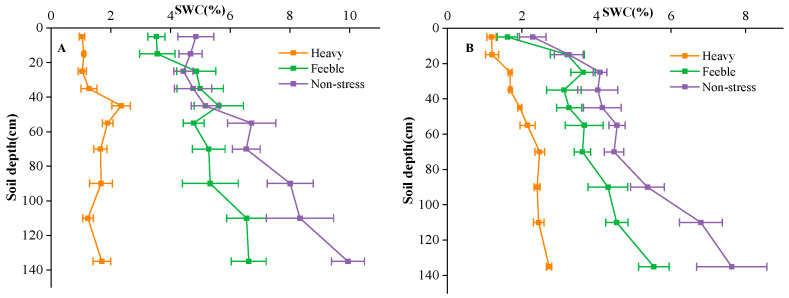
Soil water content (SWC) of florescence stage (**A**) and seed stage (**B**) in different fields. Heavy means heavy drought stress; feeble means feeble drought stress; non-stress means no drought stress. Error bars represent standard deviation. The same below.

**Table 1 plants-13-02006-t001:** Leaf morphological traits of *A. canescens* under drought stress.

	Florescence Stage			Seed Stage		
	Heavy	Feeble	Non-Stress	Heavy	Feeble	Non-Stress
LL (10^−2^ m)	2.65 ± 0.61 c	4.56 ± 0.52 b	5.67 ± 0.73 a	2.87 ± 0.20 c	4.62 ± 0.35 b	5.12 ± 0.52 a
LW (10^−2^ m)	0.50 ± 0.049 b	0.68 ± 0.068 a	0.78 ± 0.12 a	0.48 ± 0.027 b	0.68 ± 0.069 a	0.72 ± 0.082 a
LT (10^−2^ m)	0.044 ± 0.0019 b	0.049 ± 0.0023 a	0.047 ± 0.0040 ab	0.043 ± 0.0033 b	0.048 ± 0.0042 a	0.050 ± 0.0022 a
LA (10^−4^ m^2^)	0.92 ± 0.22 c	2.25 ± 0.37 b	3.06 ± 0.86 a	0.90 ± 0.073 c	1.78 ± 0.31 b	2.13 ± 0.27 a
LI	5.34 ± 1. 02 b	6.68 ± 0.86 a	7.31 ± 0.79 a	5.97 ± 0.49 b	6.88 ± 0.67 a	7.18 ± 0.81 a
FW (10^−3^ g)	51.09 ± 21.33 b	112.134 ± 22.99 a	127.42 ± 28.31 a	40.23 ± 7.74 c	62.39 ± 4.84 b	78.75 ± 17.68 a
SFW (10^−3^ g)	66.43 ± 16.77 b	159.36 ± 28.25 a	188.60 ± 55.01 a	49.94 ± 11.59 c	73.26 ± 5.21 b	91.91 ± 20.91 a
DW (10^−3^ g)	12.93 ± 4.25 b	30.10 ± 5.18 a	32.36 ± 8.95 a	13.48 ± 4.40 b	18.75 ± 2.33 ab	22.91 ± 7.06 a
SLA (10^−4^ m^2^ g^−1^)	73.68 ± 10.59 b	76.36 ± 14.89 b	94.86 ± 9.52 a	73.00 ± 20.31 b	97.16 ± 25.95 a	101.31 ± 27.36 a
LD (10^6^ g m^−3^)	0.31 ± 0.038 a	0.28 ± 0.043 a	0.23 ± 0.0243 b	0.35 ± 0.13 a	0.22 ± 0.040 b	0.22 ± 0.082b
LDMC (%)	19.20 ± 2.46 a	18.91 ± 1.27 a	17.21 ± 0.82 a	26.57 ± 3.43 a	25.61 ± 2.69 a	24.56 ± 2.15 a
SWC (%)	80.80 ± 2.46 a	81.09 ± 1.27 a	82.79 ± 0.82 a	73.43 ± 3.42 a	74.39 ± 2.69 a	75.44 ± 2.15 a
WC (%)	73.50 ± 3.59 a	72.84 ± 2.65 a	74.13 ± 5.30 a	67.17 ± 5.03 a	69.90 ± 3.24 a	71.30 ± 2.95 a

Heavy means heavy drought stress, feeble means feeble drought stress, non-stress means no drought stress. LL: leaf length; LW: leaf width; LT: leaf thickness; LA: leaf area; LI: leaf index; FW: fresh weight; SFW: saturated fresh weight; DW: dry weight; SLA: specific leaf area; LD: leaf density; LDMC: leaf dry matter content; SWC: saturated water content; WC: water content. The data are shown in mean ± SD. Different letters after the data (a, b and c) in the same indicators at the same stage mean significant difference (*p* ≤ 0.05). The same below.

## Data Availability

Data are available in a publicly accessible repository.

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
