# Peer review of "Effects of Drought Stress on Leaf Functional Traits and Biomass Characteristics of Atriplex canescens"

_plants, 2024, doi:10.3390/plants13142006_

Round 1

Reviewer 1 Report

Comments and Suggestions for Authors

Abstract

- Please highlight the most significant results in the summary by including specific values, such as percentages or other relevant metrics

Introduction

- Line 78 : As mentioned for the first time in the introduction section, please provide the datail scientific name of the species being studied : “Atriplex  Canescens” instead of “A. Canescens”

- Line 84 : Could you mention other interests of the species that highlight its high economic value?

Materials and Methods

- For example lines 123 and 129 : (1200 m3 hm-1 and 900 m3 hm-1) please review the style of the numbers and signs and put them in superscript; general remark for the entire manuscript

- Line 127 : “… (Apr., Jun., Aug., and Oct.)”, please add “and”

- Lines 140-141: “ (…G0109W, G0111W, and G0501W, respectively)”, kindly add “and”

- Line 175 : “p 0.05” instead of “p < 0.05”, general remark for the entire manuscript

Results

- Line 205 and 248 : Please ensure that figures 2 and 4 include indications of the statistical analysis results, similar to those mentioned in the titles of figure 1 and table 3.

- Line 297: Please ensure that the title of the figure 5 is as detailed as that of figure 4.

Discussion

- Kindly avoid referring to figures in the discussion section (for example :line 314, 325, …)

Author Response

Abstract

Comment 1:  Please highlight the most significant results in the summary by including specific values, such as percentages or other relevant metrics

Response: Thank you for your suggestion. We have revised the biomass and biomass allocation part in abstract to highlight the changes of biomass and biomass partitioning patterns to adapted to the drought stress. (Lines 21-24.)

Total biomass decreased 60% to 1758 g under heavy drought stress and the seed and leaf biomass was only 10% and 20% of non-stress group, but there had no significant difference on root biomass. More biomass was allocated to root under drought stress. The root biomass allocation ratio was doubled from 9.62% to 19.81% under heavy drought, and the root shoot ratio (R/S) increased from 0.11 to 0.25. (lines 21-25)

Introduction

Comment 2: Line 78: As mentioned for the first time in the introduction section, please provide the detail scientific name of the species being studied: “Atriplex Canescens” instead of “A. Canescens

Response: Thank you for your suggestion. We have used “Atriplex Canescens” instead of “A. Canescens”. (Line 79)

Comment 3: Line 84: Could you mention other interests of the species that highlight its high economic value?

Response: Thank you for your suggestion. We have added the utilization of A. canescens in roadside slope protection and mining sites restoration. We have also mentioned the potential exploitation of A. canescens in ultra-high altitude regions. (lines 84-88)

In the recent decade, A. canescens has been widely used in railway and highway side slope protection and mining sites ecological restoration. A. canescens grows well in high altitude areas of Qinghai Tibet Plateau at 4200 m (Cuomei County, Tibet autonomous region), and its ecological and feed utilization value is being highly valued by local government in some ultra-high-altitude regions. (lines 84-88)

Materials and Methods

Comment 4: For example, lines 123 and 129: (1200 m3 hm-1 and 900 m3 hm-1) please review the style of the numbers and signs and put them in superscript; general remark for the entire manuscript

Response: Thank you for your suggestion. We have checked and revised the style of the numbers and signs of the entire manuscript. The revised numbers and signs are Shown in lines 129,135 and 235(Table 1).

Comment 5: Line 127: “… (Apr., Jun., Aug., and Oct.)”, please add “and”

Response: Thank you for your suggestion. We have corrected the error. (line 133)

Comment 6: Lines 140-141: “(…G0109W, G0111W, and G0501W, respectively)”, kindly add “and”

Response: Thank you for your suggestion. We have corrected the error. (line 153)

Comment 7:  Line 175: “p ≤ 0.05” instead of “p < 0.05”, general remark for the entire manuscript

Response: Thank you for your suggestion. We have revised it as you recommendation. We have checked and revised it as you recommendation of the entire manuscript. The revised substances are shown in lines 188, 202, 221, 242, 264, 303 and 317.

Results

Comment 8:  Line 205 and 248: Please ensure that figures 2 and 4 include indications of the statistical analysis results, similar to those mentioned in the titles of figure 1 and table 3.

Response: Thank you for your suggestion. We have added the related statements about the indications of the statistical analysis results. The revised contents are Shown in lines 220-221 (original figures 2, newly figure 3), 262-265(original figures 3, newly figure 4),300(original figures 4, newly figure 5)

Comment 9:  Line 297: Please ensure that the title of the figure 5 is as detailed as that of figure 4.

Response: Thank you for your suggestion. We have added the related statements about the indications of the statistical analysis results. The revised contents are Shown in lines 316-318  (original figures 5, newly figure 6).

Discussion

Comment 10: Kindly avoid referring to figures in the discussion section (for example: line 314, 325, …)

Response: Thank you for your suggestion. We have deleted the referred figures in discussion to increase the coherence and readability of the article. These problems are shown in original manuscript of line 315,325,340,432,439,450, and there have no referred figures in discussion in the revised version.

Reviewer 2 Report

Comments and Suggestions for Authors

attached file

Author Response

Comment 1: This manuscript aims to provide information on the performance and adaptive mechanism of Atriplex canescens under drought stress by measuring the leaf functional traits and biomass characteristics, as three years of experiments with three drought gradients (heavy drought, feebly drought and non-drought stress) were performed in a common garden. However, in this research, the plants are not in controlled conditions and not only drought stress occurs:

- Moreover, the authors mentioned that: “This probably because of the experiment at seed stage was conducted in early Oct., at that time the temperature in the common garden was low (the night temperature was near 5℃), the A. canescens suffered from the combined stress of drought and low temperature.” (lines 342-345)

Response: Thank you for your comment on our study. The experimental was conducted in a common garden. The only difference among the three experimental fields is the frequency of irrigation. Except the weeds mowed annually, no other management practice was applied in the common garden during the experimental time. Therefore, we are confident that the plants are grew under controlled conditions.

As you mentioned and we have descried in section 2.2 and lines 365-370(Revised Manuscript), part of the data collected in early October, the Atriplex canescens suffered the stress of low temperature. The data collected at the seed stage is indeed affected by low temperature stress, but low temperature affected the three experimental fields equally, and the only difference of the three experimental fields was still the drought gradient in that time. Throughout our results, the changes of those leaf physiological traits shown same variation tendency between heavy drought group and non-stress group. To sum up, the low temperature stress at seed stage has no obvious effect on our research topic of the performance of A. canescens under drought stress and how A. canescens adapted to drought stress.

While, we still believe that your comment is very important, and we have further explained this situation in the article in section 4.1 after the discussion of indicators changes in seed stage to answer any doubts that readers may have. The relative contents as below.

Although the low temperature in seed stage increased the activity of antioxidant enzymes activity and the content of osmotic substances, the difference of those indicators under drought gradients has shown the same pattern with florescence stage. Therefore, the results of the seed stage further verified the physiological adaptation mechanism of A. canescens to drought. (lines 371-376)

Comment 2: Also, the soil moisture (water content) should be measured and included in Section 2.2.

Response: Thank you for your suggestion. The soil moisture has been measured at the same time with the leaf functional traits measurement. Water content has been included in section 2.2.(lines138-139,142-146)

Comment 3: Although the leaf physiological characteristics are related mainly to photosynthesis (It is mentioned in the Introduction, see also Reference 16), it is not measured here, and the photosynthetic efficiency cannot be mentioned in the conclusion (see lines 463-464). Here, the authors rather measured the biochemical characteristics/traits of the plants (see https://doi.org/10.1590/S1677-04202012000400007).

Response: Thank you for your comment. We have carefully studied the literature you mentioned and

more relevant literature on physiological and biochemical traits. Following a comprehensive review, we used “biochemical traits” instead of “physiological traits” where the contents are related to SPAD, MDA, oxidant enzyme and osmotic adjustment substance. Which are shown in the lines 55-57,149,160,193,299,301,321,330,358,361,365,463.

We fully agree with you that the relationship we discussed in discussion section of these indicators with the photosynthetic efficiency cannot be used as a conclusion, so we have deleted the relevant contents in the conclusion.

Thank you again for your suggestion. We will pay extra attention to the relevant content on physiological and biochemical traits in future research and article writing.

Comment 4: Reference should be given to the statement in the sentence "Under drought stress, A. canescens reduced leaf thickness, which may improve photosynthetic efficiency" (lines 363-369).

Response: Thank you for your suggestion. As you mentioned in comment 6, if there are no statistically difference, we cannot claim that there is a decrease or increase. Followed this criterion, we thought the discussion about leaf thickness are meaningless, so we deleted related contents in manuscript.  

Comment 5: Reference should be given for all used methods in Section 2.

Response: Thank you for your suggestion. We have added reference for the methods in section 2.3. Which are shown in lines 153, 156, 166-167,174,176,183.

Comment 6: When describing the results, it should be considered whether there are statistically different changes between non-stressed and stressed plants. The authors cannot claim that there is a decrease or an increase in some parameters if there are no statistical differences, as different small letters indicate statistical differences. For example, values for MDA are not statistically different from the controls (Figure 1, Fluorescence and seed stage), and SPAD, Pro, SS values are different only for heavy stress. Also in Table 1, the values for LDMC (%) are not statistically different, etc. Therefore, the text should be corrected in Results and Discussion. Also, the conclusions that MDA increases under drought stress in the Abstract and Discussion do not fit the results (Fig. 1).

Response: Thank you for your suggestion. We fully agree with you that if there is no statistically difference, we cannot claim there had a change and shown it in abstract and collusion. We have carefully checked and corrected the relative contents in Abstract, Results, Discussion and Conclusion. Which are shown in lines17, 20, 194-196, 210-211, 215-216, 228-229, 231, 233-234, 341-343, 353-354, 377-380, 386.

Comment 7: The conclusions section should not be an extension of the discussion. The conclusions section should illustrate the mechanistic links of findings obtained under applied treatments. The authors should avoid repeating what has already been presented in the discussion. Instead, in conclusion the mechanisms of plant adaptation to different drought stresses should be more precisely and clearly mentioned. A scheme or summary diagram can be made here to show better how A. canescens responds to two types of drought stress, i.e. to explain adaptation strategies.

Response: Thank you for your very insightful suggestion. We have revised our Conclusion to show how A. canescens responds to drought and adapted to drought. The related contents as below.

Our common garden experiment results found that the biochemical and morphological traits of leaves, as well as the biomass and biomass allocation of A. canescens significantly changed under drought stress. The POD and CAT activity decreased, the content of osmotic adjustment substance increased and the content of MDA and activity of SOD maintained. Under drought stress, leaves became smaller but denser and maintained high water content and stable dry matter content. Drought stress significantly decreased the total biomass and all organs biomass except the root. Furthermore, the biomass allocated to leaves and reproductive organs decreased, the root allocation rate increased, and the root shoot ratio increased. In general, under drought stress, A. canescens maintains survival through biochemical regulation and a tradeoff between rapid growth and drought resistance was made based on leaf morphology, biomass and biomass allocation, and a conservative survival strategy has been adapted. Specifically, A. canescens increased its investment into roots to absorb the limited water and reduced its investment into leaves and offspring to reduce the consume of water, in order to adapt to arid environments. (lines 485-499)

Round 2

Reviewer 2 Report

Comments and Suggestions for Authors

Although the revision has improved the manuscript to some extent, some corrections still need to be made before the manuscript can be accepted for publication:

Some sentences in the Abstract need correction and clarification: 

- “A. canescens maintained the content of malondialdehyde (MDA) and the activity of superoxide dismutase (SOD), but the peroxidase (POD) and catalase (CAT) activity decreased and the content of Proline (Pro) and soluble sugar (SS).”  - should be: “A. canescens maintained the content of malondialdehyde (MDA) and the activity of superoxide dismutase (SOD), but the peroxidase (POD) and catalase (CAT) activity decreased, and the content of Proline (Pro) and soluble sugar (SS) increased only under heavy drought stress.” (or similar according to the results obtained - see lines: 212-216).

- “The MDA was significantly and positively correlated with biomass, …” - It is not clear here, how MDA significantly and positively correlated with biomass, since MDA does not change but the total biomass decreases by 60%? –(see on lines 17 and 21).  Perhaps the authors should also check their results in Section 3.4.

Author Response

Comment 1: Although the revision has improved the manuscript to some extent, some corrections still need to be made before the manuscript can be accepted for publication:

Some sentences in the Abstract need correction and clarification: 

- “A. canescens maintained the content of malondialdehyde (MDA) and the activity of superoxide dismutase (SOD), but the peroxidase (POD) and catalase (CAT) activity decreased and the content of Proline (Pro) and soluble sugar (SS).”  - should be: “A. canescens maintained the content of malondialdehyde (MDA) and the activity of superoxide dismutase (SOD), but the peroxidase (POD) and catalase (CAT) activity decreased, and the content of Proline (Pro) and soluble sugar (SS) increased only under heavy drought stress.” (or similar according to the results obtained - see lines: 212-216).

Response:Thank you for your recognition of our revised manuscript, and once again thank you for providing new suggestions to make our manuscript more specialty and professional. We revised the sentence aligned with your recommendation. The revised sentence as below.

  1. canescens maintained the content of malondialdehyde (MDA) and the activity of superoxide dismutase (SOD), but the peroxidase (POD) and catalase (CAT) activity decreased, and the content of Proline (Pro) and soluble sugar (SS) increased only under heavy drought stress. (lines:17-19)

Comment 2:- “The MDA was significantly and positively correlated with biomass, …” - It is not clear here, how MDA significantly and positively correlated with biomass, since MDA does not change but the total biomass decreases by 60%? – (see on lines 17 and 21).  Perhaps the authors should also check their results in Section 3.4.

Response:Thank you for your comment. We have checked our results and discussion, and we found we made a mistake when we summarized about the positive and negative correlation of MDA and SPAD to biomass in the abstract. We corrected the sentence in the abstract as below and there are no related problems in Results and Discussion section in newly revised manuscript.

The MDA was significantly and negatively correlated with biomass, while the SPAD was significantly and positively correlated with total and aboveground organs biomass. (Lines: 25-27)

We have noticed the doubt of the reviewer about why MDA does not change but it has a significant correlation with the significantly decreased biomass. This because of even there have no significant difference in MDA content under drought stress, but the increase of MDA content under heavy drought should not be ignored. We didn’t emphasize and gives prominence of this changes followed the reviewer’s comment 6 in round 1, which suggested if there is no statistically difference, we cannot claim there had a change. Under the prolonged drought stress, the persistent high content of MDA reflected the water limitation damaged the biomembrane system, which affected the normal physiological and biochemical function and inhibited the growth of A. canescens.